# A Transformer Model for Symbolic Regression towards Scientific Discovery

**Florian Lalande** [*]
OMRON SINIC X Corporation
Okinawa Institute of Science and Technology
`florian.lalande@oist.jp`

**Yoshitomo Matsubara** [†]
Amazon Alexa
`yomtsub@amazon.com`

**Naoya Chiba**
Tohoku University
`chiba@nchiba.net`

**Tatsunori Taniai**
OMRON SINIC X Corporation
`tatsunori.taniai@sinicx.com`

**Ryo Igarashi**
OMRON SINIC X Corporation
`ryo.igarashi@sinicx.com`

**Yoshitaka Ushiku**
OMRON SINIC X Corporation
`yoshitaka.ushiku@sinicx.com`

## Abstract

Symbolic Regression (SR) searches for mathematical expressions which best describe numerical datasets. This allows to circumvent interpretation issues inherent to artificial neural networks, but SR algorithms are often computationally expensive. This work proposes a new Transformer model aiming at Symbolic Regression particularly focused on its application for Scientific Discovery. We propose three encoder architectures with increasing flexibility but at the cost of column-permutation equivariance violation. Training results indicate that the most flexible architecture is required to prevent from overfitting. Once trained, we apply our best model to the SRSD datasets (Symbolic Regression for Scientific Discovery datasets) which yields state-of-the-art results using the normalized tree-based edit distance, at no extra computational cost.

## 1 Introduction

Machine Learning methods allow to infer relationships in datasets and replicate them, but they are often criticized for being black-box models, preventing from understanding the assumptions and discoveries of the model. Symbolic Regression (SR) aims at addressing this problem by searching the space of mathematical expressions and find an interpretable analytical model to explain a given dataset. SR originated in the field of Genetic Programming [1], and most state-of-the-art methods still use this approach. However, Genetic Programming algorithms tend to be computationally expensive, and deep learning methods have recently attracted attention from the SR community due of their almost instantaneous inference [2–10]. Notably, the "End-to-End Symbolic Regression with Transformer" model of Kamienny et al. [8] provides close to state-of-the-art performances (when using the $R^2$ accuracy metric) at virtually no extra computational time.

Because of its interpretability, SR has been extensively applied in various scientific domains. A pioneer study by Schmidt and Lipson [11] proposed to automatically rediscover physical laws by

---

[*]This work was done while the first author was a research intern at OMRON SINIC X Corporation.
[†]This work was partly done while the second author was a research intern at OMRON SINIC X Corporation.

NeurIPS 2023 AI for Science Workshop.

only providing experimental data on the position and velocity of objects. Since then, the sub-field of Symbolic Regression for Scientific Discovery (SRSD) tries to draw the line between SR seen as an exercise for Machine Learning research on one side, and its potential for automated or assisted scientific discovery on the other.

As the SR community was growing, La Cava et al. [12] proposed SRBench, a modern, living, and unified framework to benchmark SR methods. However, although they used physically-motivated equations, Matsubara et al. [13] point out that existing SR datasets have several important shortcomings: unrealistic sampling process, inappropriate handling of physical constants (e.g. the speed of light is sampled in the range $[1, 5]$), lack of diversity in orders of magnitude, the systematic treatment of integer variables as continuous variables, or the fact only relevant variables are provided in the input dataset. Another problem of SR tasks lies in the lack of meaningful evaluation metrics specifically for SRSD. To tackle these issues, Matsubara et al. [13] propose SRSD datasets based on the 120 equations from the Feynman SR datasets [14], meticulously reviewing and highlighting properties for each dataset, and split into three categories: 30 easy, 40 medium, and 50 hard datasets. They also propose the use of the normalized tree-based edit distance to better assess how structurally close from the ground-truth the predicted equations are.

This work introduces a new Transformer model tailored for SR in the context of scientific discovery. We propose three encoder architectures and train our model using generated datasets. We then evaluate our best model on the SRSD datasets using the normalized tree edit distance. We show that our best model achieves state-of-the-art results on the SRSD datasets at almost immediate inference time. We also publish our code repository for future studies[1].

## 2 Methodology – Symbolic Regression with Transformers

This section introduces the architecture for our Transformer model, the methodology used to generate the training dataset, and the adopted training strategy.

### 2.1 Architecture of our Transformer model

Our model is a Transformer model [15] adapted to the problem of SR (see Figure 1). It comprises an encoder, which receives a numerical tabular dataset as input, and a decoder, whose task is to predict a sequence of tokens corresponding to the ground-truth equation. The role of the encoder is to convert the tabular dataset of numerical values into meaningful features. We propose and analyze the performances of three architectures for the encoder.

We keep in mind that the features of the numerical tabular datasets (i.e. the output of the encoder) are expected to be permutation invariant with respect to the rows (the observations), and permutation equivariant with respect to the columns (the variables). On one hand, row-permutation invariance requires that the features do not change when the rows of the tabular dataset are shuffled. On the other hand, column-permutation equivariance calls for similar changes between the input and the output. For example, if the ground-truth equation is $\log(x_1 - x_2)$ and the variables $x_1$ and $x_2$ are switched in the original tabular dataset, we expect the features to change accordingly such that our model should predict $\log(x_2 - x_1)$.

The encoder of our Transformer model starts with a Cell MLP, followed by a stack of $N_{\text{enc.}}$ identical layers, where we propose three architectures for the encoder layers (see items below, and architecture details in Appendix A). A final MLP layer followed by MaxPooling allows to build features that are permutation invariant with respect to the rows.

- `MLP` **encoder layer** is made of a MLP, a row-wise Max-Pooling, another MLP, and a column-wise Max-Pooling. This architecture preserves both row-permutation invariance and the column-permutation equivariance.
- `Att` **encoder layer** uses a multi-head self-attention mechanism layer between variable features. Like the `MLP` architecture, this encoder layer type preserves both the row-wise permutation invariance and the column-wise permutation equivariance properties.
- `Mix` **encoder layer** starts by blending the column features using a standard MLP layer. This layer is inspired by PointNets [16], an artificial neural network architecture satisfying

---

[1] https://github.com/omron-sinicx/transformer4sr

row-permutation invariance and taking point clouds as input. The output of this first layer is then passed to a multi-head self-attention mechanism layer to look at other observations (rows) in the numerical dataset. While the row-permutation invariance is preserved, the MLP operation breaks the column-permutation equivariance property.

The decoder of our model first combines (learnable) token embeddings and (fixed) token positional encodings. The main part of the decoder is composed of $N_{\text{dec.}}$ stacked identical layers. Following the typical architecture for the decoder of the Transformer model, each layer has: (i) a masked multi-head self-attention layer, (ii) another multi-head attention layer where the queries and the keys come from the output of the encoder and the values come from the previous layer, and (iii) a final standard MLP layer. An additional MLP layer outputs a matrix of dimension $(M \times v)$, where $M$ is the maximum number of tokens allowed and $v$ is the vocabulary size. As a Transformer model, the decoder of our model is auto-regressive, i.e. the first $n$ outputs of the decoder are fed back into the decoder itself to produce output $n + 1$ during inference.

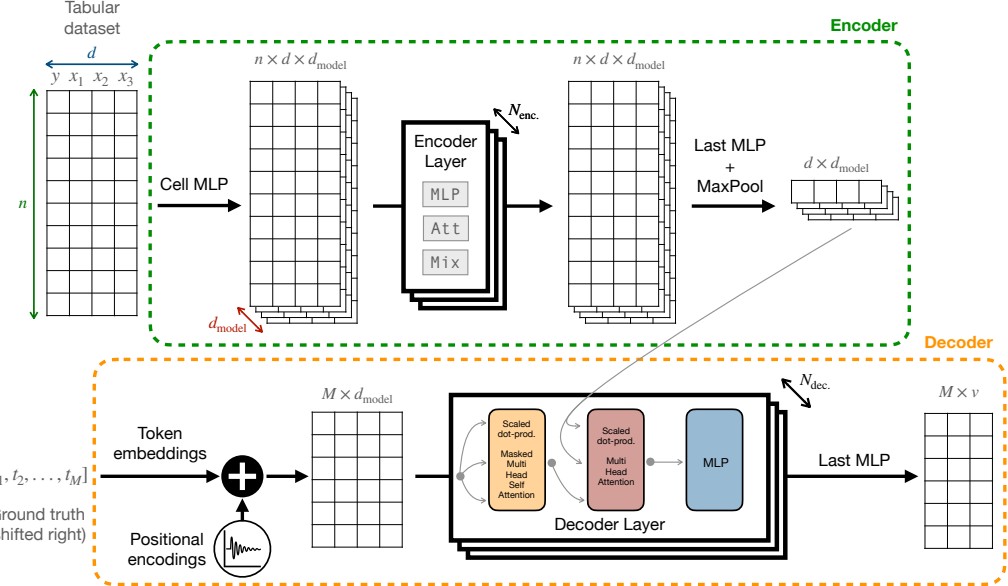

Figure 1: **Architecture of our Transformer model for Symbolic Regression.** We propose three encoder architectures: `MLP`, `Att`, or `Mix`. The decoder is a standard Transformer decoder and is the same in all cases. During training, the encoder receives the tabular dataset and the decoder receives the ground-truth sequence of tokens, used with teacher-forcing method. During inference, the decoder is on its own and predicts tokens in an auto-regressive manner.

We introduce $d_{\text{model}}$, the innermost dimension of our model, which corresponds to the dimension of the feature space used for internal representations. After few trials and errors, and given our computational limitations, we fix $d_{\text{model}} = 256$ as well as $N_{\text{enc.}} = 4$, $N_{\text{dec.}} = 8$, and the number of independent heads for the multi-head attention layers is $h = 4$. Given these hyperparameters, the `MLP`, `Att`, and `Mix` models respectively have $6,863,892$, $7,523,348$, and $9,622,548$ trainable parameters (more on that in Appendix A).

## 2.2 Generating synthetic training datasets

The model is trained using synthetic tabular datasets, aimed at representing the diversity of real world data typically collected during scientific experiments. We first generate a large variety of skeleton equations using `SymPy` [17] which we then sample from.

We begin by generating ground-truth equations, by sampling tokens from a fixed predefined vocabulary composed of the following tokens: [`add, mul, sin, cos, log, exp, neg, inv, sq, cb, sqrt, C, x1, x2, x3, x4, x5, x6`]. Tokens `sq` and `cb` respectively denote squared and cubed values. Token `C` denotes a constant whose value will be later sampled. Expressions can include

up to six variables, denoted from $x_1$ to $x_6$. Each token is given a sampling weight (see the code) to account for typical frequencies of operators, e.g. `mul` is more common than `cos`. Note that we do not include the binary operations corresponding to division and subtraction, but instead use `inv` and `neg` to represent the unary operation of inverse and negative.

Equations are represented as trees in `SymPy`. The two tokens `add` and `mul` represent binary operators and correspond to binary nodes requiring two children. Tokens `sin`, `cos`, `log`, `exp`, `neg`, `inv`, `sq`, `cb`, and `sqrt` are unary operators, represented as nodes needing one child. The remaining tokens `C`, `x1`, `x2`, `x3`, `x4`, `x5`, and `x6` represent constant or variables, and correspond to the tree leaves. We start by sampling a token from the vocabulary, and continue the sampling process until the tree is completed, i.e. when all nodes have all their children.

We generate $N_{\text{sample}} = 1,000,000$ random equations. These equations are simplified in a consistent way using `SymPy`. Besides, we discard equations made of a single leaf, equations that do not make use of the constant $C$ in their skeleton, and equations that do not have a single variable. We also discard expressions that have more than 30 tokens. We are left with $112,944$ valid expressions, from which we finally discard duplicates and end up with $17,438$ unique valid expressions. For each remaining equation, the ground-truth is stored as a sequence of tokens from the equation tree, using the pre-order traversal algorithm allowing to maintain a one-to-one correspondence between equation tree and sequence of tokens. For example, the skeleton of the equation $y = 10x_1 + x_2 \log(x_1)$ will be stored as the sequence `[add, mul, C, x1, mul, x2, log, x1]`.

For each unique valid expression, we uniformly sample a value for the constant $C$ in the range $[-100; +100]$, and we log-uniformly sample 50 times the values for the variables $x_1, ..., x_K$ (where $K \leq 6$) in the range $[10^{-1}; 10^1]$. The log-uniform range for the variables is chosen to represent the natural variability of physical phenomena. This sampling process generates a tabular dataset of size $(50, 7)$, where 50 is the number of observations and 7 corresponds to the response variable followed by the maximum number of variables allowed $(y, x_1, x_2, x_3, x_4, x_5, x_6)$, possibly padded with zeros.

The whole sampling process is repeated 100 times, such that there exist several tabular datasets corresponding to the same ground-truth, but with different values for the constants and the variables. Lastly, we discard datasets from which sampling was impossible (e.g. $\log(-x_1 x_2)$, because $x_i \geq 0$) or datasets where $|y| > 10^9$. We are left with $1,494,588$ tabular datasets for training.

## 2.3 Model training

Our model is fed numerical tabular datasets and is tasked to output tokens. In the beginning, we initiate the decoder with the start of sequence `<SOS>` token. Besides, the ground-truth sequences are padded with the `<PAD>` token, so that all ground-truths have size $M = 31$ (including the `<SOS>` token). Our vocabulary of tokens is the same as the one used for equation generation (see previous subsection) with the addition of the `<SOS>` and `<PAD>` tokens. The vocabulary size is $v = 20$.

We use the categorical cross-entropy loss function with the Teacher Forcing method during training. We also test with and without label-smoothing for the loss function (label-smoothing of $\epsilon = 0.1$) when training, and compare results in the next section.

Following the default parameters of Vaswani et al. [15], the parameters of our Transformer model are updated with the Adam optimizer [18], with an initial learning rate $\gamma = 1$, decay factors of $\beta_1 = 0.9$ and $\beta_2 = 0.98$ and small scalar $\epsilon = 10^{-9}$. Besides, we use the same learning rate scheduler as [15] with $4,000$ warm-up steps followed by an inverse square-root decay. During training, we monitor the loss function as well as the token-wise accuracy (seen as a $v$-class classification problem). Finally, we use dropout regularization with $p_{\text{drop}} = 0.25$.

The entire training dataset is split into train, validation, and test subsets with proportions $80\%$, $10\%$ and $10\%$ respectively. We provide batches of size $1,024$, which corresponds to $1,168$ training iterations to complete an epoch. We train for 100 epochs using 4 NVIDIA Tesla V100 GPUs with 16GB memory. One epoch takes about 8 min to be completed, such that the whole training process was completed in approximately 13 hours.

# 3 Results

This section first introduces the results regarding training our Transformer model with respect to the proposed encoder architectures and training loss functions, and then presents the performances of our best model on the SRSD datasets [13].

## 3.1 Training results with respect to encoder architectures

We present the training results obtained for different training scenarios. Figures 2 and 3 show the evolution of the loss function and the token-wise accuracy ($v$-class classification) with respect to training epochs for all three proposed architectures. Figure 2 corresponds to the case without label-smoothing and Figure 3 to the case with label-smoothing $\epsilon = 0.1$.

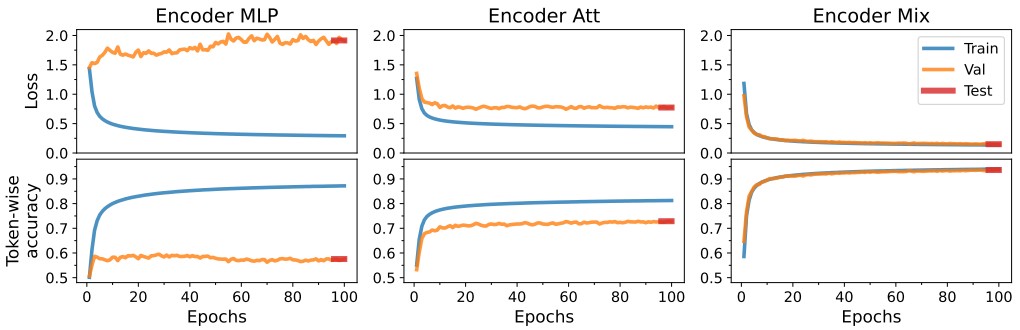

Figure 2: **Loss function and token-wise accuracy during training without label-smoothing.** The MLP encoder architecture strongly overfits the training set and cannot generalize to the validation/test sets. The Att encoder architecture can somehow generalize to the validation/test sets but still shows some overfitting. The Mix architecture shows no overfit sign at all.

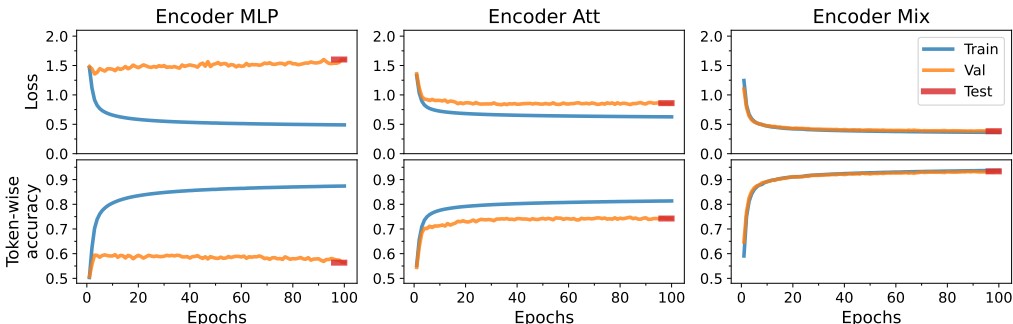

Figure 3: **Loss function and token-wise accuracy during training with $\epsilon = 0.1$ label-smoothing.** The same statement as Figure 2 applies. Label-smoothing does not resolve the overfitting problem.

As can be seen on Figure 2, the MLP and Att encoder architectures overfit the training set and do not generalize their learning to the validation and test sets. The overfit is much stronger for the MLP architecture. In contrast, the Mix encoder architecture do not show any sign of overfitting, and both the loss function and the token-wise accuracy computed on the validation or test sets match with the train set. The validation and the test sets are statistically equivalent, and therefore exhibit similar performances at epoch 100 because we do not use early-stopping with the validation set.

An intuitive justification for this state of affairs is that the Mix architecture allows for arbitrary interactions between the variables $x_1, ..., x_6$ and $y$, by blending them together. The subsequent self-attention layer of the Mix encoder allows to look for relations between observation-wise features and therefore build an internal representation of the whole tabular dataset. On the other hand, the MLP and Att architectures dedicate most of their parameters to computing variable-only features.

Their encoder crosses the information between variables only through few MaxPooling operations (for the `MLP` architecture) or self-attention layers (for the `Att` architecture). In other words, trying to preserve the permutation equivariance property with respect to variables is too restrictive and hinders the flexibility of the encoder, while violating this property with the `Mix` architecture instead allows for building more meaningful features.

Introducing label-smoothing does not change the relative behaviour of the three proposed architectures. However, we can see on Figure 3 that the loss function plateaus at higher values (which makes sense because of label-smoothing), while the token-wise accuracy appears to be unaffected.

## 3.2 Performances on SRSD datasets

Now, we move to the evaluation of our best Transformer model using the SRSD datasets proposed to evaluate SR methods tailored for scientific purposes. The set of the 120 SRSD problems is available online[2] and is split into three subsets: 30 easy, 40 medium, and 50 hard equations [13].

Each equation in the SRSD dataset is comprised of $10,000$ observations, divided into $8,000$ training, $1,000$ validation, and $1,000$ test observations. As our model is already pre-trained using the generated training dataset presented in Section 2.2, we only use the $1,000$ observations from the test set for each equation in the SRSD dataset.

There exist several heterogeneous metrics to assess the performances of SR algorithms. Notably, La Cava et al. [12] propose to use three metrics in their open-source benchmark, called SRBench. These are (i) the $R^2$ test for the accuracy, (ii) the formula complexity for the interpretability, and (iii) the inference time for the rapidity. However, these metrics do not assess how structurally close to the ground-truth equation the model prediction is, and are mostly useful when the ground-truth equations are unknown.

For this reason, and because the ground-truth equations are available, we decide to focus on the estimation of the correct skeleton equation, considered to be the hardest problem. We use the normalized tree-based edit distance proposed by Matsubara et al. [13] defined as:

$$\tilde{d}(f_{\text{pred}}; f_{\text{true}}) = \min\left(1; \frac{d(f_{\text{pred}}; f_{\text{true}})}{|f_{\text{true}}|}\right)$$

where $d(f_{\text{pred}}; f_{\text{true}})$ is the tree edit distance computed with the Zhang and Shasha algorithm [19] between the predicted $f_{\text{pred}}$ and the ground-truth $f_{\text{true}}$ equations represented as trees, and $|f_{\text{true}}|$ is the number of tokens (or tree nodes) for the ground-truth equation. It has been shown that this metric is more aligned with human judgment than the $R^2$ scores in cases where the ground-truth is known [13].

During evaluation, we start with formatting the SRSD datasets such that the variables $x_1$ to $x_6$ lie in the range $[10^{-1}; 10^1]$ to match the range of the generated datasets used during training. We also log-normalize the response variable $y$ and compensate it by adding an extra token `C` in the ground-truth to take the scaling factor into account. Because there exist several ways of expressing the same mathematical expression, we standardize the ground-truth equations using the `simplify` and `factor` SymPy functions, which are the same conventions used during training (see Section 4 for details). Finally, recall that our model takes as input tabular numerical datasets of size $(50, 7)$. Therefore, we randomly sample $N = 50$ valid observations from the $1,000$ available observations in the test sets, and we repeat this sampling process 30 times for each SRSD dataset.

Table 1 presents the mean of the normalized tree-edit distance for each SRSD dataset category and each of our six trained models. These results indicate that the `Mix` encoder architecture provides best solutions when applied to the SRSD datasets. The use of label-smoothing in the loss function does not seem to significantly affect the performances of our model, except maybe on the easy SRSD datasets. Therefore, we select the `Mix` Transformer model trained with label-smoothing as our best model.

## 3.3 Comparison with state-of-the-art Symbolic Regression methods

We compare these results with the performances of six baseline SR algorithms used in [13], where five of the baselines are state-of-the-art SR methods, according to the SRBench study [12].

---

[2]`huggingface.co/datasets/yoshitomo-matsubara/srsd-feynman_{easy;medium;hard}`

|        | No label smoothing | | | With label smoothing | | |
|--------|------|------|------|------|------|------|
|        | MLP | Att | Mix | MLP | Att | Mix |
| **easy** | 0.975 | 0.821 | 0.740 | 0.896 | 0.896 | 0.686 |
| **medium** | 0.938 | 0.795 | 0.678 | 0.885 | 0.799 | 0.697 |
| **hard** | 0.857 | 0.778 | 0.732 | 0.840 | 0.800 | 0.747 |

Table 1: **Normalized tree edit distance for the proposed Transformer models.** The `Mix` encoder architecture provides best results on the SRSD datasets, regardless of the use of label-smoothing in the loss function.

- `gplearn`, a Genetic Programming Python library built with Scikit-Learn [20].
- Age-Fitness Pareto (AFP), a Genetic Programming method using Pareto optimization which takes into account the model's age (epoch) when training [21].
- AFP-FE, which corresponds to AFP using Eureqa for fitness estimation [11]. As a commercial platform, Eureqa cannot be easily integrated in the benchmark.
- AI-Feynman (AIF), a physics-driven SR algorithm using successive divide and conquer fixed rules. They also introduce the Feynman datasets [14].
- Deep Symbolic Regression (DSR), using recurrent neural networks [3].
- End-to-End Symbolic Regression with Transformer (E2E), another Transformer-based model considered state-of-the-art for SR [8]. They use the scientific notation for tokens even within the encoder, unlike our model which uses the raw numerical values.

Table 2 presents the results of our best model (Best m.) against other traditional baselines for SR tasks, where the scores have been taken from [13]. We can see that our best Transformer model provides the lowest normalized tree-edit distance results among all methods when evaluated on the SRSD medium and hard datasets. For the easy SRSD datasets, DSR and AI-Feynman achieve the best solutions.

|        | gplearn | AFP | AFP-FE | AIF | DSR | E2E | Best m. |
|--------|---------|-----|--------|-----|-----|-----|---------|
| **easy** | 0.876 | 0.703 | 0.712 | 0.646 | 0.551 | 1.000 | 0.686 |
| **medium** | 0.939 | 0.873 | 0.897 | 0.936 | 0.789 | 1.000 | 0.697 |
| **hard** | 0.978 | 0.960 | 0.956 | 0.930 | 0.833 | 0.981 | 0.747 |

Table 2: **Aggregated performances on the SRSD datasets.** Our best Transformer model (`Mix` encoder with label-smoothing) outperforms other traditional SR methods on the medium and hard SRSD datasets, and provides competitive performances on the easy SRSD datasets.

It is worth mentioning that our Transformer model being already trained, the inference is almost instantaneous at test time, unlike the first five SR algorithms. Therefore, it benefits from the same advantage as the End-to-End Transformer for (E2E) SR proposed by Kamienny et al. [8] while providing better results on unseen datasets coming from various scientific fields.

## 4 Discussion

Because of the token-wise accuracy close to 93% during training (see Figures 2 and 3), we might expect even lower normalized tree-edit distance results. However, the normalized edit distance results during evaluation with the SRSD datasets are worse than what they would be if computed over the generated tabular numerical datasets used for training. This is because of at least two reasons. The first one is that the Transformer model is now on its own in an auto-regressive manner during inference, as the Teacher Forcing method cannot be used during test on the SRSD datasets. This means that early errors get propagated and amplified during decoding. The second one is that as we increase the ground-truth equation complexity (typically its number of tokens) there is exponentially more and more possible equations. As a result, the medium and hard SRSD equations most likely do not appear in the generated datasets used during training. And while some equations of the easy

SRSD datasets might be in the training set, typical equations with small length (less than 10 tokens) are underrepresented in the training set, which leads our model to often output more than necessary tokens.

This observation leads to the discussion of an important problem when training Transformer models tailored for SR: the difference between in-domain performance versus out-of-domain generalization. While one can easily overfit the training set with enough parameters and computing time, this does not guarantee that the Transformer model can extrapolate its learning to out-of-domain datasets. When they introduced their "End-to-End Symbolic Regression with Transformers" (E2E), Kamienny et al. [8] explained that memorization of the training set does not occur based on a statistical reasoning (see Appendix C of [8]). We would instead argue that memorization does occur, and there is no point trying to prevent this from happening. Memorization at least guarantees that the tabular datasets presented to the Transformer during training will be correctly identified during inference at test time. The actual hurdles instead include: building diverse and representative tabular datasets, handling constants and variables frequency, choosing appropriate sampling ranges, or finding ways to represent ground-truth equations coherently – e.g. which of $y = (x_1 - x_2)^2$, $y = x_1^2 + x_2^2 - 2x_1 x_2$, or $y = x_2^2 - 2x_1 x_2 + x_1^2$ should the Transformer model predict, although all correct?

As most traditional SR algorithms can be computationally expensive, Transformer models take advantage of prior GPU training to allow for almost instantaneous inference results later on. However, further improvements and new research should still be done as to what architectures and training strategies are optimal. In particular, we showed here that trying to preserve the permutation invariance and equivariance properties might be unreasonably restrictive constraints, potentially hindering the flexibility of the artificial neural network models. We also note that depending on the chosen metric(s), ranking of methods can greatly vary. While SRBench [12] discuss the performance of SR methods focused on the $R^2$, the model complexity, and the inference time, we decide to use the normalized edit distance as suggested by Matsubara et al. [13] as this metric allows to quantify how structurally close to the ground-truth the estimation is. It should be an important metric to discuss the potential of symbolic regression for scientific discovery as the metric considers both the interpretability and structural similarity between the true and predicted equations.

## 5   Conclusion

Because of the vast searching space, Symbolic Regression (SR) is a very complicated problem, and its real-world application towards scientific discovery is even more complex. Now, a major strength of Transformer models for SR is that their inference is almost instantaneous, because they have already been trained for long hours beforehand. In this work, we proposed several Transformer model architectures tailored for SR and aiming to solve automatic scientific discovery tasks. Our best model demonstrates strong performances on the SRSD datasets, a large dataset for SR in the context of scientific discovery. It already improves over the results of E2E, another Transformer model for SR, and there is still much room for improvement. In particular, generating more diverse training datasets, allowing for more flexibility with the constants, having a more flexible vocabulary of allowed tokens, or bigger/more refined Transformer architectures could improve our proposed Transformer model. Our code is open-sourced and can be accessed at the following repository[1].

## Acknowledgments and Disclosure of Funding

We used the computational resources of AI Bridging Cloud Infrastructure (ABCI) provided by the National Institute of Advanced Industrial Science and Technology (AIST). This work was supported by JSTMirai Program Grant Number JPMJMI21G2 and JST Moonshot R&D Grant Number JPMJMS2236, Japan. We would also like to thank the anonymous reviewers and meta-reviewer for their insightful comments and suggestions for our work.

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

# A   Proposed encoder architectures

This appendix provides additional information regarding the proposed three encoder architectures and their properties. Figures 4, 5, and 6 respectively represent the `MLP`, the `Att`, and the `Mix` encoder architectures.

The `MLP` encoder architecture is the first we propose, and is shown on Figure 4. This encoder architecture has desirable properties regarding numerical tabular datasets. The "Cell MLP" allows to preserve the row permutation invariance and column permutation equivariance. Dataset-wise features are constructed using the "Row MaxPool" and the "Column MaxPool" operations. Note that the innermost dimension (the dimension of the Cell MLP) has been divided by two in order to conserve the same dimension after concatenations with the MaxPooling outputs. While this architecture preserves interesting properties, it shows the worst results during training and evaluation on the SRSD datasets. This is because most of the `MLP` encoder parameters are dedicated to construct cell representations, but only few of them allow for interactions between variables (the columns of the dataset).

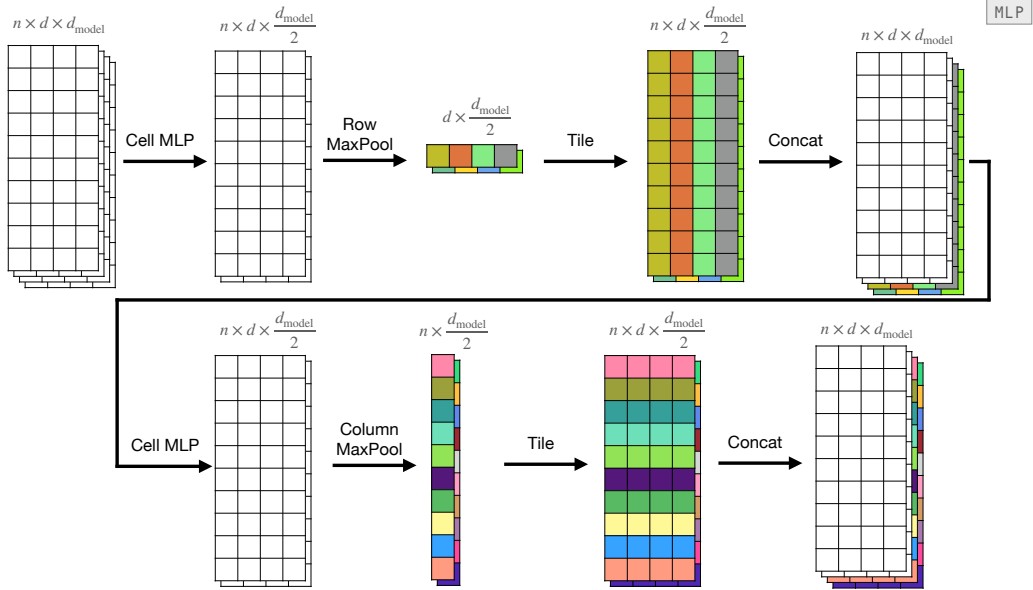

Figure 4: **Encoder architecture – `MLP` version.** This encoder architecture preserves row permutation invariance and column permutation equivariance. After MaxPooling, the features are tiled and concatenated to the original tensor. This architecture does not allow for much flexibility in feature design between variables.

In light of the shortcomings of the `MLP` encoder architecture, we then propose the `Att` encoder architecture (c.f. Figure 5) which utilizes multi-head self-attention mechanisms to build more flexible features from the input tabular dataset. The `Att` encoder architecture conserves the same properties as the previous encoder architecture, because self-attention layers do not break the row permutation invariance or the column permutation equivariance when applied on the innermost dimension (the feature dimension, see Figure 5). This means that the self-attention mechanism is applied on a cell basis and across variables, i.e. seeking relevant information in other columns (other variables) but not across different samples. As could be seen in Section 3, this architecture also allows for better generalization: the validation and test loss/accuracy are closer to the train loss/accuracy with the `Att` encoder than with the `MLP` encoder. However, we can still observe some overfitting and the small gap between train and validation or test metrics indicate that there is additional room for improvement.

Finally, the `Mix` encoder architecture offers the most flexible design at the expense of violating the column permutation equivariance properties. This architecture has been inspired by PointNets, an artificial neural network architecture to work with point clouds classification and segmentation tasks [16]. As can be seen on Figure 6, the `Mix` encoder architecture starts by flattening the columns features altogether, such that original 3D feature tensors become 2D feature tensors. The following

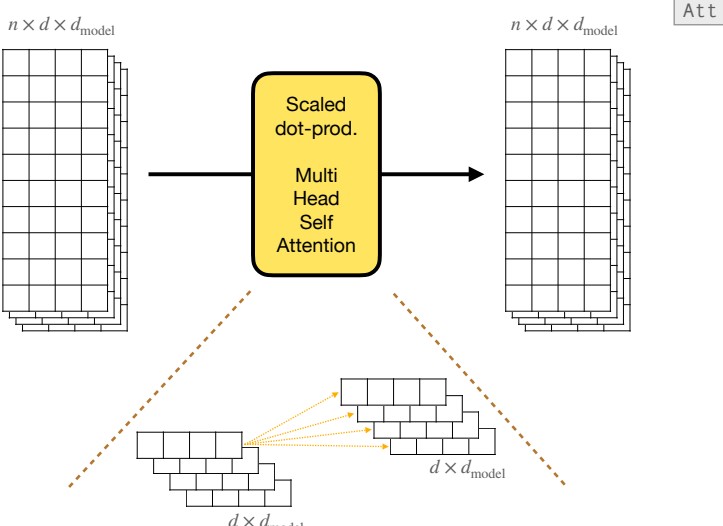

Figure 5: **Encoder architecture – `Att` version.** The self-attention mechanism preserves row permutation invariance and column permutation equivariance, and is more flexible in feature design.

MLP is not a "Cell MLP" anymore, but instead blends information from all columns: this allows for arbitrary relations between the input variables $x_k$ and the response variable $y$. Next, a multi-head self-attention mechanism layer is applied onto the $(n \times d_{\mathrm{model}})$ tensor, which allows to compare and match sample features across the whole dataset. This step preserves the row permutation invariance and allows for greater flexibility in feature design. Finally, we add and normalize (residual connection) the original dataset with its new features (after being broadcasted), which restores the original shape.

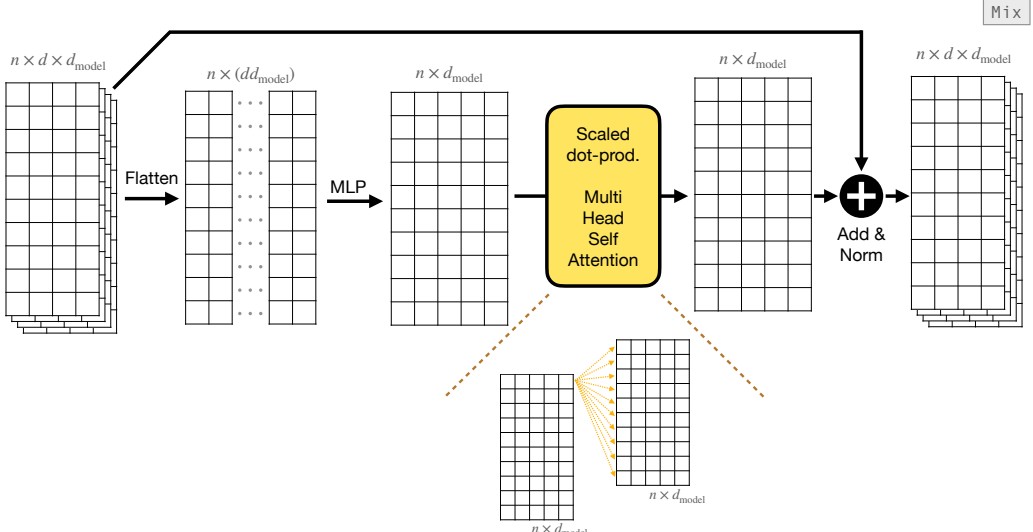

Figure 6: **Encoder architecture – `Mix` version.** This architecture violates column permutation equivariance, but allows for the computation of arbitrary relationships between variables.

During the review process of this manuscript, it has been brought to our attention that column-permutation equivariance could still be enforced during training by using data-augmentation strategies. Indeed, for each equation in our synthetic training dataset, we could draw random column permutations and change the ground truth accordingly. For example, suppose the ground-truth equation is given by $y = \log(x_1) + x_1 x_2$, we could then swap columns 2 and 3 (corresponding to $x_1$ and $x_2$ respectively) in the tabular dataset and convert the ground-truth to $\log(x_2) + x_2 x_1$ accordingly.

For the three proposed architectures, the number of trainable parameters varies depending on the chosen hyper-parameters. Table 3 presents the number of parameters for each building block.

| | | | Number of parameters |
|---|---|---|---|
| **Encoder** | **First Cell MLP** | | $d_{\mathrm{model}}^2 + 3d_{\mathrm{model}}$ |
| | **One Encoder Layer** | **MLP** | $\frac{3}{2}d_{\mathrm{model}}^2 + 2d_{\mathrm{model}}$ |
| | | **Att** | $4d_{\mathrm{model}}^2 + 6d_{\mathrm{model}}$ |
| | | **Mix** | $(5+d)d_{\mathrm{model}}^2 + 8d_{\mathrm{model}}$ |
| | **Last MLP** | | $d_{\mathrm{model}}^2 + d_{\mathrm{model}}$ |
| **Decoder** | **Token Embeddings** | | $vd_{\mathrm{model}}$ |
| | **One Decoder Layer** | | $12d_{\mathrm{model}}^2 + 17d_{\mathrm{model}}$ |
| | **Last MLP** | | $vd_{\mathrm{model}} + v$ |

Table 3: **Number of trainable parameters per building block.** The overall number of trainable parameters scales quadratically with $d_{\mathrm{model}}$, the innermost dimension of our Transformer model. For the `Mix` encoder architecture, the number of trainable parameters also scales linearly with the number of columns $d$ in the tabular datasets. Finally for the decoder, the number of trainable parameters depends on the vocabulary size $v$.

Note that for the encoder and decoder parts, the number of trainable parameters is given for a single layer. They should be multiplied by the number of encoder layers $N_{\mathrm{enc.}}$ and decoder layers $N_{\mathrm{dec.}}$ before being added to obtain the total number of trainable parameters $N_{\theta,\mathtt{xxx}}$ for each architecture.

$$N_{\theta,\mathtt{MLP}} = (1.5N_{\mathrm{enc.}} + 12N_{\mathrm{dec.}} + 2)d_{\mathrm{model}}^2 + (2N_{\mathrm{enc.}} + 17N_{\mathrm{dec.}} + 2v + 4)d_{\mathrm{model}} + v$$

$$N_{\theta,\mathtt{Att}} = (4N_{\mathrm{enc.}} + 12N_{\mathrm{dec.}} + 2)d_{\mathrm{model}}^2 + (6N_{\mathrm{enc.}} + 17N_{\mathrm{dec.}} + 2v + 4)d_{\mathrm{model}} + v$$

$$N_{\theta,\mathtt{Mix}} = ((5+d)N_{\mathrm{enc.}} + 12N_{\mathrm{dec.}} + 2)d_{\mathrm{model}}^2 + (8N_{\mathrm{enc.}} + 17N_{\mathrm{dec.}} + 2v + 4)d_{\mathrm{model}} + v$$

As a sanity check, we can plug the values used in this work, i.e. $N_{\mathrm{enc.}} = 4$, $N_{\mathrm{dec.}} = 8$, $d = 7$, $v = 20$, and $d_{\mathrm{model}} = 256$. We can verify that $N_{\theta,\mathtt{MLP}} = 6,863,892$, $N_{\theta,\mathtt{Att}} = 7,523,348$, and $N_{\theta,\mathtt{Mix}} = 9,622,548$, as introduced at the end of Section 2.1 in the main text.

