# OpenReview forum: "A Transformer Model for Symbolic Regression towards Scientific Discovery"
_NeurIPS.cc/2023/Workshop/AI4Science — NeurIPS2023-AI4Science Oral_

### Official Review · Reviewer_yuL3 · 2023-10-24
**Comments on "A Transformer Model for Symbolic Regression towards Scientific Discovery"**

**Rating:** 9
**Confidence:** 4

**Review:**

This manuscript addresses a very interesting and important topic: symbolic regression. The authors proposed a new framework and conducted comparative studies for several cases. It's a solid study, and I enjoy reading it very much. Everything is clear to me except for Figure 2 and Figure 3.

For these two figures, the errors for testing and validation dataset almost share the same error with the training dataset, which is a big surprise to me. It would be helpful if the authors could provide more analysis on this and maybe some extra tests to ensure there is nothing wrong with the results.

---

### Official Review · Reviewer_1fEZ · 2023-10-24
**Strong analysis, thorough experimentation, and very fruitful insights**

**Rating:** 9
**Confidence:** 4

**Review:**

I really appreciate your contribution.

The SOTA on SRSD is a great bi-product of your analysis and thorough investigation, and your discussions about the research is SR at the end of the paper is quite meaningful.

One question: Can you try to add data augmentation to the Mix architecture to encourage equivariance to the order of the variables? It may be an interesting way to reconcile the lack of equivariance in a good parameterization (which is what your experiments suggest to my understanding).

Missing references on SR for scientific discovery:

https://ieeexplore.ieee.org/document/9180100
https://arxiv.org/abs/2007.10784
https://arxiv.org/abs/2210.00563

---

### Meta-Review · Area_Chair_KSos · 2023-10-27

**Recommendation:** Accept (Oral)
**Confidence:** 5

**Metareview:**

**Overview:**
The paper presents a novel Transformer model tailored for Symbolic Regression (SR), with a keen focus on aiding scientific discovery. The authors delve into three proposed encoder architectures, emphasizing their varying degrees of flexibility and the trade-offs with column-permutation equivariance. The paper establishes its novelty through its impressive performance on the SRSD datasets, marking state-of-the-art results. Additionally, the work is complemented by rigorous analysis and experimentation, further substantiating its claims.

**Strengths:**

1. **Novel Approach:** The introduction of a Transformer model specifically for SR, especially within the realm of scientific discovery, marks a significant departure from traditional methods, thereby adding considerable novelty to the paper.

2. **Thorough Analysis:** The depth of analysis, particularly the consideration of flexibility against equivariance in encoder architectures, is commendable. Such granularity enhances the paper's rigor and credibility.

3. **Superior Performance:** Achieving state-of-the-art results on the SRSD datasets not only validates the efficacy of the proposed model but also highlights its potential for broader applications.

4. **Insightful Discussions:** The insightful discussions around SR at the conclusion of the paper amplify its significance and contextualize its contributions within a larger academic discourse.

**Constructive Feedback from Reviewers and Areas for Enhancement:**

1. **Data Augmentation Consideration:** Reviewer 1fEZ suggests the exploration of data augmentation, specifically for the Mix architecture. This could introduce additional robustness to the model and further enrich the study.

2. **Clarification on Figures:** Reviewer yuL3 points out potential anomalies in Figures 2 and 3, where the testing, validation, and training datasets exhibit nearly identical errors. This calls for a more detailed examination, validation, and, if necessary, clarification or amendment of these results to ensure authenticity.

3. **Missing References:** To ensure the paper's comprehensiveness and academic integrity, it's crucial to incorporate the additional references on SR for scientific discovery, as highlighted by Reviewer 1fEZ.

**Recommendation:**
Given the paper's original contributions, its alignment with the current research trajectory, and its evident strengths, it is highly recommended for an oral presentation. The paper not only offers innovative solutions but also engages deeply with existing challenges, making it ideal for discussion and discourse in a conference setting. Addressing the constructive feedback provided by the reviewers will further solidify its position as a top-tier contribution to the field.